# Speech emotion recognition using machine learning techniques: Feature extraction and comparison of convolutional neural network and random forest

**Mohammad Mahdi Rezapour Mashhadi**[1]*, **Kofi Osei-Bonsu**[2]

**1** Independent Researcher, Mashhad, Iran, **2** Ghana Data Center, Accra, Ghana

* mohammadmashhadi3188@gmail.com

**Data Availability Statement:** Various data types were used and all are publicly available based on the data section.

## Abstract

Speech is a direct and rich way of transmitting information and emotions from one point to another. In this study, we aimed to classify different emotions in speech using various audio features and machine learning models. We extracted various types of audio features such as Mel-frequency cepstral coefficients, chromogram, Mel-scale spectrogram, spectral contrast feature, Tonnetz representation and zero-crossing rate. We used a limited dataset of speech emotion recognition (SER) and augmented it with additional audios. In addition, In contrast to many previous studies, we combined all audio files together before conducting our analysis. We compared the performance of two models: one-dimensional convolutional neural network (conv1D) and random forest (RF), with RF-based feature selection. Our results showed that RF with feature selection achieved higher average accuracy (69%) than conv1D and had the highest precision for fear (72%) and the highest recall for calm (84%). Our study demonstrates the effectiveness of RF with feature selection for speech emotion classification using a limited dataset. We found for both algorithms, anger is misclassified mostly with happy, disgust with sad and neutral, and fear with sad. This could be due to the similarity of some acoustic features between these emotions, such as pitch, intensity, and tempo.

## Introduction

Emotions are an essential part of human communication and interaction. They can convey information about the mental state, intention, attitude and personality of a person. However, emotions are not always easy to recognize and interpret, especially when they are expressed through speech. People may express the same emotion in different ways, using different tones, pitches, intensities and words. Moreover, some emotions may be subtle or mixed, making them hard to distinguish even for human listeners.

Analyzing the human voice and facial expressions to automatically recognize emotions has attracted a lot of interest [1, 2]. Various methods and techniques have been proposed to extract

**Funding:** The authors received no specific funding for this work.

**Competing interests:** There is no competing interest

and classify emotions from speech and visual signals. Speech emotion recognition (SER) is the task of automatically identifying and classifying the emotional state of a speaker from their speech signal, regardless of the semantic content. SER has many potential applications in various security [3], entertainment [4], detecting depression [5], and monitoring students' engagement [6].

SER is a challenging machine learning problem that involves extracting relevant features from audio signals and applying suitable models to classify them into different emotion categories. There are many factors that can affect the performance of SER systems, such as the quality and diversity of the audio data, the choice and representation of the emotion classes, the selection and extraction of the audio features and the design and evaluation of the models.

Audio classification is a more general machine learning task that involves identifying and tagging audio signals into different classes and groups. Audio classification can have many applications such as anomaly detection [7], fault detection [8], health monitoring [9] or music genre recognition. Audio classification can also be seen as a prerequisite or a subtask for SER, as it can help to filter out irrelevant or noisy audio signals that do not contain emotional information.

The majority of past studies that evaluated audio for emotion recognition considered audios individually which might be biased due to invariability of the dataset. For instance, an ensemble 1D-CNN-LSTM-GRU was used [10], where the model achieved a promising results while considering a single dataset at a time, For instance, the model achieved 95.42% for TESS, 95.62 for RAVDESS, and 93.22% for SAVEE datasets. Or for instance, [11] applied their hybrid CNN- LSTM models on 6 different datasets separately. They achieved, 90.42% accuracy for RADVDESS, or 99.48% for the TESS datasets. In contrast, our approach combines multiple datasets to provide a more comprehensive evaluation of audio emotion recognition.

In this paper, we present a comparative study of two machine learning models for SER: one-dimensional convolutional neural network (conv1D) and random forest (RF) with RF-based feature selection. We use a limited dataset of SER, consisting of eight emotion classes: calm, happy, sad, angry, fearful, disgust, surprised and neutral. We augment the dataset with additional audio to increase its size and diversity.

We extract various types of audio features from the speech signals such as Mel-frequency cepstral coefficients (MFCC), chromagram (CHROMA), mel-scale spectrogram (MEL), spectral contrast feature (SCF), Tonnetz representation (TONNETZ) and zero-crossing rate (ZCR). We compare the performance of conv1D and RF with feature selection on the augmented dataset using various metrics such as accuracy, precision and total accuracy.

Some of the highlights of this paper are:

- We apply a method of augmenting a limited SER dataset with additional audios using techniques such as pitch shifting and time stretching.

- We compare the performance of conv1D and RF with feature selection on the augmented dataset using six types of audio features.

- We show that RF with feature selection outperforms conv1D on the augmented dataset and achieves higher average accuracy (69%) than conv1D.

- We analyze the results and discuss the strengths and limitations of each model and feature type.

- Our approach combines multiple datasets to provide a more comprehensive evaluation of audio emotion recognition.

## Dataset

We used four different sources of publicly-available audio datasets that contain speech samples of various actors expressing different emotions. The following paragraphs provide a brief description of each dataset. For the majority of audios, we noticed that the first and the last one second of the audio file had no audio signal in the wave shape. Therefore, we trimmed this one and last second from the audio file for all the files in the dataset to remove any silence.

### Ravdess

Ryerson Audio-Visual Database of Emotional Speech and Song (RAVDESS) is a gender-balanced database that consists of 24 professional actors in a neutral North American accent [12]. The database contains speech and song samples in 8 emotions: calm, happy, sad, angry, fearful, disgust, surprised and neutral. We used the voice-only format of the database in our analysis.

### Crema

Crema Crowd-sourced Emotional Multimodal Actors (CREMA) is a dataset of 7,442 clips from 91 actors. The clips were collected from 48 male and 43 female actors between the age of 20 and 74 across a variety of races and ethnicities. The actors spoke one of 12 sentences using one of six different emotions: anger, disgust, fear, happy, neutral and sad [13]. The sentences were chosen from the International Affective Picture System (IAPS) and the International Affective Digitized Sounds (IADS).

### Tess

Tess Toronto Emotional Speech Set (TESS) is a collection of audio samples from two female actors aged 26 and 64 years old. The actors spoke 200 target words in seven emotions: anger, disgust, fear, happiness, pleasant surprise, sadness and neutral [14]. The target words were selected from the Affective Norms for English Words (ANEW) database.

### Savee

Savee Surrey Audio-Visual Expressed Emotion (SAVEE) is a dataset of audio samples from four male actors aged between 27 and 31 years old. The actors spoke 15 sentences in seven emotions: anger, disgust, fear, happiness, sadness, surprise and neutral [15]. The sentences were taken from the TIMIT corpus and modified to suit different emotions. The data were recorded in a visual media lab and processed and labeled. It is important to note that all of the datasets used in our study are publicly available and can be accessed through the references provided for each dataset.

Various characteristics of audios were presented in Fig 1. In the left figure, the figure computed short-time Fourier transform (STFT) of the audio data. The right figure shows short-time Fourier transform (STFT) of the audio data. The STFT is then converted to decibels. It could be seen that the differences between emotions such as anger and sadness are not easily distinguishable in the waveplot and spectrogram figures.

Emotions are complex and can be expressed in many different ways, and it can be challenging to accurately classify emotions based on audio data alone. That is evident from the visual representations of audio data, such as waveplots and spectrograms, which may not always clearly show the differences between emotions.

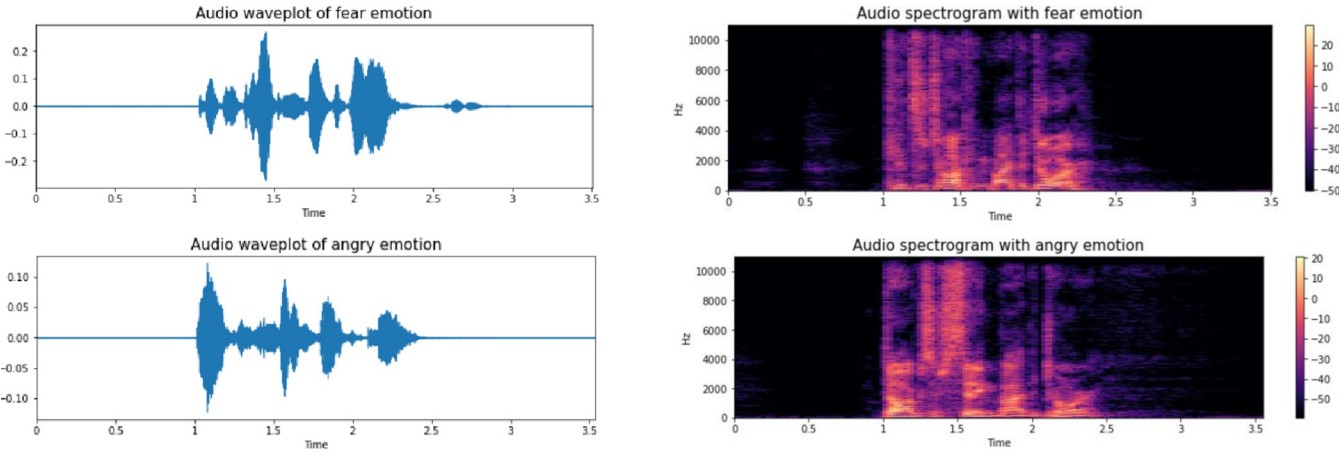

**Fig 1. Audio waveplot and spectrogram of fear and anger emotions.**

## Method

In this section, we describe the steps we followed to preprocess and augment the audio data, extract and select the audio features, design and train the conv1D and RF models, and compare and analyze the results.

### Data preprocessing and augmentation

We used four datasets of speech emotion recognition: RAVDESS [10], CREMA [11], TESS [12] and SAVEE [13]. Each dataset contained speech samples of different actors expressing different emotions. We used eight emotion classes: calm, happy, sad, angry, fearful, disgust, surprised and neutral. All audios were converted to a common sampling rate of 16 kHz and a common bit depth of 16 bits.

To increase the size and diversity of our data, we augmented each audio file using techniques: adding noise, pitch shifting and time stretching. Noise defines a function that takes an audio data array as input and adds some random noise to it. The noise amplitude is proportional to the maximum value of the data and a random factor between 0 and 0.035. The noise is generated from a normal distribution with the same shape as the data array. The function returns the noisy data array as output.

Stretch defines a function called stretch that takes an audio data array and a rate factor as input and stretches or compresses the data in time. The rate factor determines how much the data is stretched or compressed. A rate factor less than 1 means that the data is stretched (slowed down), while a rate factor greater than 1 means that the data is compressed (sped up), where we used rate of 0.7.

Shift defines a function called shift that takes an audio data array as input and shifts it in time by a random amount. The shift range is determined by a random integer between -5 and 5 multiplied by 1000. The function uses the numpy.roll function to perform the circular shift of the data array. The function returns the shifted data array as output.

Pitch defines a function called pitch that takes an audio data array, a sampling rate and a pitch factor as input and changes the pitch or frequency of the data. The pitch factor determines how much the pitch is changed. A pitch factor less than 1 means that the pitch is lowered, while a pitch factor greater than 1 means that the pitch is raised. The function uses the librosa.effects.pitch_shift function to perform the pitch shifting. The function returns the pitch-shifted data array as output. We used a pitch factor of 0.7.

Audio feature extraction and selection We extracted various types of audio features from the pre-processed and augmented data: MFCC [16], Chroma [17], Mel [18], SCF [19], Tonnetz [19] and ZCR [20]. These features capture different aspects of the spectral and temporal characteristics of the speech signals that are relevant for emotion recognition. We used the librosa library in Python to compute these features. All the audio processing techniques were implemented by Librosa package in Python [21].The following sections will briefly talk about the considered processed.

## Zero crossing rate

Zero-crossing rate (ZCR) is a measure of the number of times that an audio signal crosses the zero amplitude level in a given time frame. ZCR captures the temporal information and noise level of speech that can be related to emotion expression. The process could be written as

$$zcr = \frac{1}{T-1} \sum_{t=1}^{T-1} 1_{R<0}(s_t s_{t-1}) \qquad \qquad 1$$

Where $s$ is a signal of length $T$ and $1_{R<0}$ is an indicator function.

For instance, assuming the sampling rate is 22,050 HZ, and frame size of 2,048 samples, then, frame size is 2,048/22,050 = 0.0929 seconds, and hop length is 512/22,050 = 0.0232 seconds, where 512 is the hop length in samples. Now the number for a 3 second audio is

$$\frac{audio\ length - frame\ size}{hop\ length} + 1 = \frac{3 - 0.0929}{0.0232} + 1$$

So we would have 126 column for this single value. We used the average of all and use that for our ZCR.

Again, the ZCR function returns a matrix with one row and as many columns as frames. By transposing this matrix, it becomes a matrix with one column and as many rows as frames. Then, we takes the mean value along the row axis.

## Mel-frequency cepstral coefficients

The Mel-frequency cepstral coefficient (MFCCs) was used, where MFCCs describe the overall shape of the spectral envelope of an audio signal. The process could be summarized as extraction of short-time Fourier transform (STFT) magnitude spectrogram, mapping the STFT into Mel scale by use of triangular overlapping windows and getting its log, and finally application of the discrete cosine transform (DCT) to the previous stage.

Mel-frequency cepstral coefficients (MFCC) are a representation of the short-term power spectrum of an audio signal based on a nonlinear Mel scale of frequency. MFCC are widely used for speech recognition and analysis because they mimic the human auditory system and capture the timbral and harmonic information of speech.

MFCCs are widely used in various fields, such as speech recognition, speaker identification, music information retrieval, and audio compression. They capture the spectral envelope, the shape of the power spectrum of sound. It describes the variation of energy across different frequency bands of the sound and discard some irrelevant details, such as pitch and noise. They can also be combined with other features, such as delta and delta-delta coefficients, which represent the changes in MFCCs over time. A frequency measured in Hertz (f) can be converted to the mel scale using the following formula:

$$Mel(f) = 295 \log_{10}\left(1 + \frac{f}{700}\right) \qquad \qquad 2$$

### Roll-off frequency

The roll-off frequency for each frame is defined as the center frequency for a spectrogram bin where at least most of (85%) the spectrum energy of the frame spectrum is contained in the bin and the bins below. The technique is useful in distinguishing audios with different energy distributions.

In other words, roll-off frequency is a feature that measures the frequency below which a certain percentage of the total energy of the spectrum is contained. It can be used to distinguish between harmonic and noisy sounds, as harmonic sounds tend to have lower roll-off frequencies than noisy sounds. Roll-off frequency can also be used for emotion classification, as different emotions may have different spectral distributions.

Some studies have used roll-off frequency as one of the features for speech emotion recognition, along with other features such as MFCCs, pitch, and energy. It does not account for the temporal dynamics and prosodic variations of speech. Therefore, it may make sense to use roll-off frequency in combination with other features and methods for emotion classification.

The equation for roll-off frequency is

$$R = \underset{k}{argmin} \left( \sum_{i=0}^{k} s(i) \geq roll_{percent} \cdot \sum_{i=0}^{n} S(i) \right) \qquad 3$$

The equation works by finding the smallest value of $k$ such that the sum of the power spectrum from 0 to $k$ is greater than or equal to a certain percentage of the total sum of the power spectrum from 0 to $n$. This percentage is called roll_percent and it is usually set to 0.85. The value of $k$ corresponds to the frequency bin that contains the roll-off frequency. The roll-off frequency is the frequency below which most of the energy of the spectrum is concentrated.

### Spectral contrast

Spectral contrast feature is based on the past study which used the technique for music type classification, which present relative spectral distribution, instead of average spectral envelop [22]. Although the technique was employed in music type classification in the past study here, we used that for the emotions associated with each audio. Librosa estimated this feature by dividing spectrogram frame of $s$ into sub bands, and then estimating energy contrast by comparing mean energy in the peach energy or top quantile to that of the bottom quantile or valley energy.

Spectral contrast feature (SCF) is a representation of the spectral peak and valley structure of an audio signal based on the contrast between spectral subbands. SCF captures the spectral dynamics and texture of speech that can be related to emotion expression.

We compute the spectral contrast for each frame of the audio data and then take the mean value across all frames. The equation for computing spectral contrast is

$$C(k) = 10.log_{10}\left(\frac{V(k)}{P(k)}\right) \qquad 4$$

where $C(k)$ is the spectral contrast in the $k$th sub-band, $V(k)$ is the spectral valley in the $k$th sub-band, and $P(k)$ is the spectral peak in the $k$-th sub-band. The spectral valley and peak are computed by finding the minimum and maximum values of the spectrum within a specified quantile range in each sub-band. The equation works by capturing the relative energy distribution across different frequency regions of the spectrum.

## Harmonic change in tonal centroid features

This measure detect harmonic changes in audios, by projecting chroma features into a 6 dimensional space [23]. Tonnetz representation is a representation of the tonal space or harmonic relations of an audio signal based on a geometric model that maps pitches to points in a two-dimensional lattice. Tonnetz representation captures the tonal information of speech that can be related to emotion expression. It consists of six dimensions: fifth, minor third, major third, tonic, subdominant, and dominant. The equation for tonnetz is:

$$T = Q^{\mathrm{T}}C \qquad\qquad 5$$

where T is the tonnetz matrix with shape (6, t), Q is a constant matrix that maps chroma features to tonnetz features, and C is the chroma matrix with shape (12, $t$).

## Chromagram from a waveform or power spectrogram

Chroma features are powerful representation for music audio where the entire spectrum is projected into 12 bins, representing 12 distinct chroma, where "Shepard Tones" which consist of a mixture of sinusoids carrying a particular chroma were used [24]. Chroma features In the previous study chroma based audio features was carried out for dialect identification [25].

The value is a representation of the pitch content of a sound or music octave. This could be written as

$$C = \frac{F^{\mathrm{T}}S}{||F^{\mathrm{T}}S||_{\infty}} \qquad\qquad 6$$

Where $S$ is the power spectrogram, and $F$ a chroma filter bank matrix that maps each frequency bin to a chroma bin.

## Root-mean-square

The root-mean-square (RMS) value is a measure of the average energy or amplitude of a signal. It is computed by squaring each sample in a frame, taking the mean of the squares, and then taking the square root of the mean. The equation for RMS is:

$$RMS = \sqrt{\frac{1}{N}\sum_{n=0}^{N-1}x[n]^2} \qquad\qquad 7$$

$x$ is the audio signal, which is a sequence of samples, where $x[n]$ is the $n$th sample in a frame of length $N$. $N$ is the length of the frame, which is a segment of the audio signal that we want to analyze.

## Mel-scaled spectrogram

The function feature.melspectrogram uses both the STFT equation and the Mel filter bank equation to compute a mel-scaled spectrogram. It also uses the Mel scale conversion equations to construct the Mel filter bank.

The steps for calculating the function are as follows:

1- If the input is a time-domain signal $y$, compute the power spectrogram $S$, using the STFT equation:

$$S[m,k] = |\sum_{n=0}^{N-1}y[n]w[n-mH]e^{-j2\pi nkn/N}|^2 \qquad\qquad 8$$

Where $k$ is the index of frequency bin, ranging from 0 to $K$-1, where $K$ is the number of

frequency bins. Where the number of frequency bins is equal to 1-n_fft/2, where n_fft is the FFT window size. The FFT window size is the number of samples used to compute the discrete Fourier transform (DFT) of a segment of the signal. It is also called the frame length or the analysis window length. The FFT window size determines the frequency resolution and the time resolution of the STFT.

$j[n]$ is the $n$th sample of the signal, $N$ is the window length, $H$ is the hop length, $w[n]$ is the window function and $j$ is the imaginary unit.

2- Now, construct Mel filter bank matrix M using the Mel scale conversion Eq 2

The Mel filter bank matrix $M$ has shape (n_mels, 1 + n_fft/2), where n_mels is the number of Mel filters. Each row of $M$ corresponds to a triangular filter that covers a certain range of frequencies on the Mel scale.

3- Apply the Mel filter bank matrix $\underline{M}$ to the power spectrogram $S$ using the Mel filter bank equation:

$$S_m[m, l] = \sum_{k=0}^{K-1} S[m, k]M[l, k] \qquad\qquad 9$$

Where $S_m[m, l]$ is the mel-scaled spectrogram coefficient at time frame $m$ and filter index $l$, and $k$ is the number of frequency bins (1-n_fft/2). The mel-scaled spectrogram S_m has shape (n_mels, t).

We concatenated these feature types into one feature vector for each audio file, resulting in a feature vector with a wide dimensions. However, not all features may be equally relevant or useful for emotion recognition. Although conv1D might do feature selection itself, RF might not be able to do so. Therefore, we applied feature selection methods to reduce the dimensionality and complexity of the feature vectors and improve the performance of the models. We used random forest-based feature selection (RFS). RFS is a method that selects a subset of features based on their importance scores derived from a random forest model.

We designed and trained two models for SER: conv1D and RF. Conv1D is a type of convolutional neural network (CNN) that applies one-dimensional convolution filters to the input feature vectors to learn local and global patterns for emotion recognition. RF is a type of ensemble learning method that combines multiple decision trees to produce a robust and accurate prediction for emotion recognition. We briefly describe each model below:

We used a conv1D model with four convolutional layers, each followed by a batch normalization layer, a rectified linear unit (ReLU) activation function, and a max pooling layer. The convolutional layers had 128 and 64 filters, respectively, with a kernel size of 5 and a stride of 1, see Table 1. The max pooling layers had a pool size of 5 and a stride of 2.

**Table 1. Conv1D architect.**

| Layer | Output shape | Other parameters |
| --- | --- | --- |
| Conv1D | 173, 128 | kernel_size = 5, strides = 1, padding = 'same', activation = 'relu' |
| MaxPooling1D | 87, 128 | pool_size = 5, strides = 2, padding = 'same' |
| Dropout | | |
| Conv1D | 87, 64 | 64, kernel_size = 5, strides = 1, padding = 'same', activation = 'relu' |
| MaxPooling1D | 44, 64 | pool_size = 5, strides = 2, padding = 'same' |
| Dropout | | |
| Flatten | 2816 | |
| Dense | 32 | units = 32, activation = 'relu' |
| Dense | 8 | units = 8, activation = 'softmax' |

After the convolutional layers, we added a flatten layer, a dropout layer with a rate of 0.5, and a dense layer with 8 units and a softmax activation function. The dense layer produced the final output of the model, which was a probability distribution over the 8 emotion classes. We used categorical cross-entropy as the loss function, Adam as the optimizer, and accuracy as the metric. We trained the model for 100 epochs with a batch size of 32, using early stopping with a patience of 10 epochs to prevent overfitting. Table 1 shows the details of the conv1D model architecture.

It is clear that the shape at each consecutive layer is estimated as follows:

$$output_{length} = (input_{length} + 2*padding - dilation*(kernel_{size} - 1)/stride + 1 \qquad 10$$

We used the default values for padding, dilation (spacing) and stride (step size) as 0, 1, and 1, respectively. As the same padding was used, it is assumed that padding is $(kernek_{size}-1)/2$, so after placing the values in the above equation we have 87, for instance, look at Table 1.

The non-linear activation function of Rectified Linear Unit (ReLU), f(x) = max(0, x) was used, due to its simplicity, avoiding vanishing gradient, and introducing sparsity to network resulting in reduction in overfitting. Softmax was used in the final layer due to its capability of converting k real numbers into k probabilities, summing up to 1.

We used an RF model with 100 decision trees, each with a maximum depth of None and a minimum samples split of 2. The RF model used the gini criterion to measure the quality of the splits, and bootstrap sampling with replacement to create the training subsets for each tree. The RF model produced the final output by averaging the predictions of all the trees, which was a probability distribution over the 8 emotion classes. We used accuracy as the metric to evaluate the performance of the model.

We split the data into three sets: training, validation and testing. We used 70% of the data for training, 15% for validation and 15% for testing. We used stratified sampling to ensure that each set had a balanced distribution of emotion classes. We used the training set to fit the models, the validation set to tune the hyperparameters and select the features, and the testing set to evaluate the performance of the models.

Few points should be highlighted regarding the choice of various values in Conv1d. The kernel size should be small enough to capture local patterns in the input but not too small to miss important info. On the other hand, the number of filters should be large enough to capture the features' diversities but not too high to result in computational inefficiency and overfitting. To prevent the possible loss of information, a lower stride value was used to preserve information but larger than 1 to address possible computational issues. Maxpooling1D works by taking the maximum value of spatial window of the size of pool size. Better performance of random forest is dependent on the data characteristics and the architects of the conv1d model. Fig 2 depicts the discussed methodological steps taken in this study.

## Results

We compared the performance of two methods for emotion recognition from audio: a convolutional neural network (CNN) with one-dimensional convolutional layers (Conv1D) and a random forest (RF) classifier with feature selection. We used a dataset of 1,440 audio clips of eight different emotions: angry, calm, disgust, fear, happy, neutral, sad, and surprise. We split the dataset into 80% training and 20% testing sets. We evaluated the methods using precision, recall, and accuracy metrics, see Table 2.

For the RF method, we first applied another RF classifier to select the most important features from the raw audio data. The performance was achieved with 100 trees, Gini impurity as a criteria for split, and no limit for maximum depth of tree. We then used these features as inputs to another RF classifier for emotion recognition.

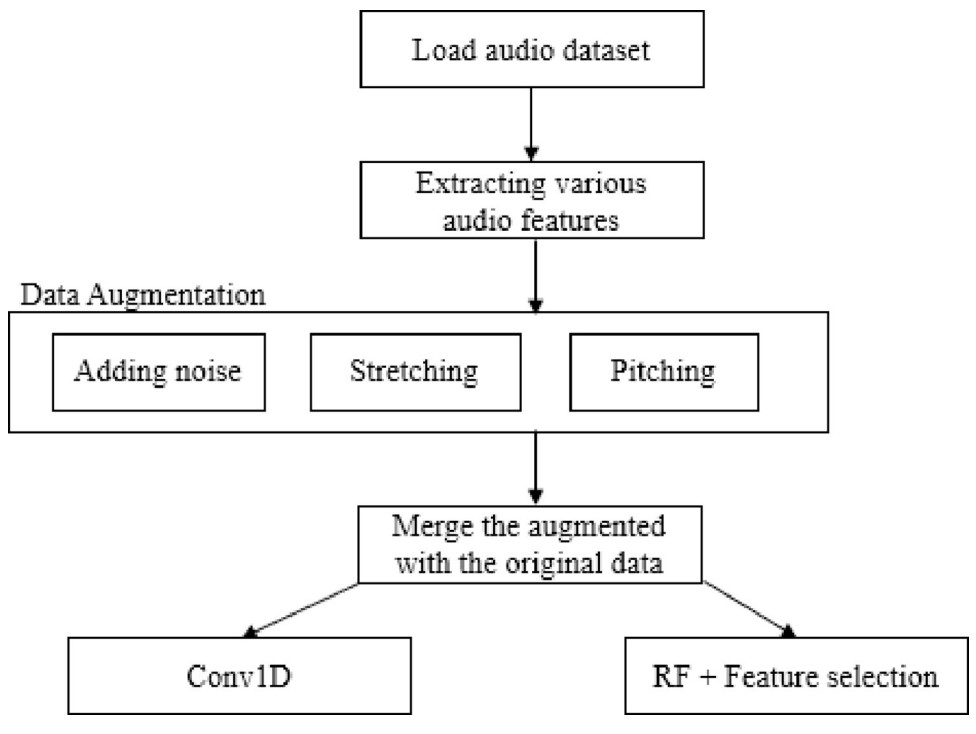

**Fig 2. Methodological steps for audio classification.**

The results show that the RF method with feature selection outperformed the Conv1D method in terms of accuracy and most of the precision and recall scores for each emotion. Originally 174 features of audios were considered. However, those columns were reduced to 93 columns by means of RFE. We found for both algorithms, angry is misclassified mostly with happy, disgust with sad and neutral, and fear with being sad.

We have included Fig 3 to provide a visual representation of the distribution of accuracy for various emotions. As can be seen from Fig 3, surprise has the best performance, while disgust has the lowest performance.

## Discussion

In this study, we explored the task of speech emotion recognition (SER) using various audio features and machine learning models. SER is a challenging task aiming to recognize and

**Table 2. Precision and recall scores for each emotion using Conv1D and RF methods.**

|  | CONV1D | | | Random forest + feature selection | | |
|---|---|---|---|---|---|---|
|  | Precision | Recall | F1-score | Precision | Recall | F1-score |
| Angry | 0.77 | 0.72 | 0.74 | 0.72 | 0.80 | 0.76 |
| Calm | 0.55 | 0.81 | 0.65 | 0.68 | 0.84 | 0.75 |
| Disgust | 0.58 | 0.45 | 0.50 | 0.62 | 0.57 | 0.60 |
| Fear | 0.69 | 0.49 | 0.57 | 0.79 | 0.55 | 0.65 |
| Happy | 0.57 | 0.57 | 0.57 | 0.64 | 0.63 | 0.64 |
| Neutral | 0.52 | 0.63 | 0.57 | 0.68 | 0.71 | 0.70 |
| Sad | 0.54 | 0.71 | 0.61 | 0.63 | 0.78 | 0.70 |
| Surprise | 0.82 | 0.85 | 0.83 | 0.85 | 0.83 | 0.84 |
| Accuracy | 0.61 | | | | | 0.69 |

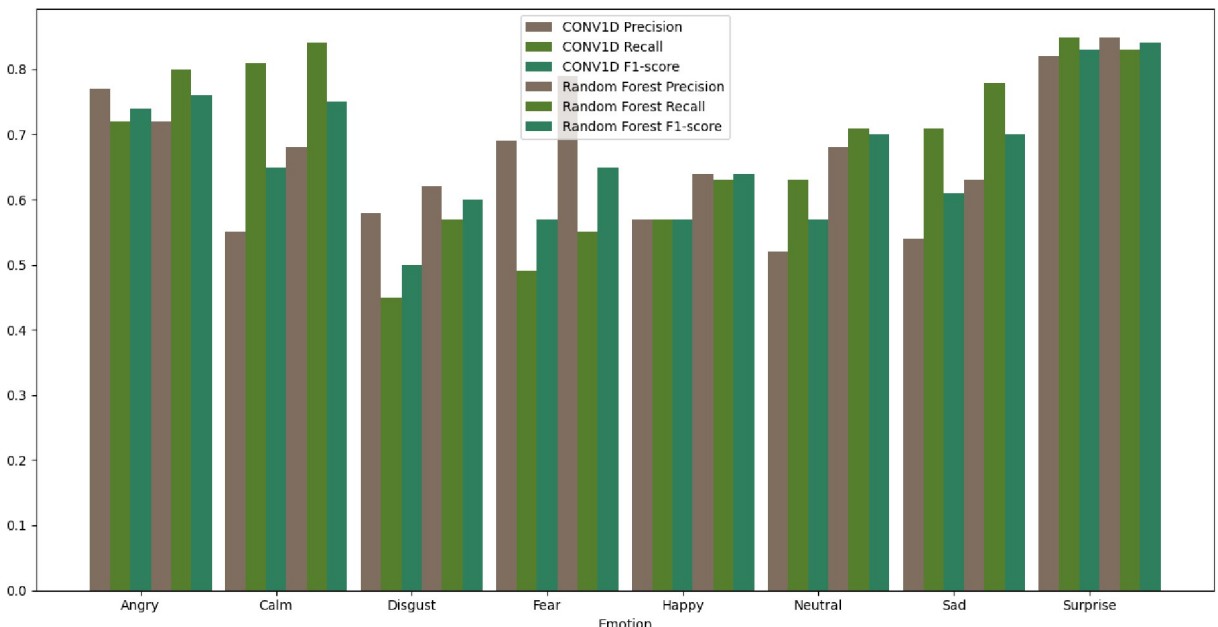

**Fig 3. Comparison of CONV1D and random forest models for audio emotion classification.**

categorize the emotions expressed in spoken language. SER can have many applications, such as interactive voice-based assistants, caller-agent conversation analysis, and mental health assessment.

We extracted various types of audio features from the raw audio data, such as Mel-frequency cepstral coefficients, chromogram, mel-scale spectrogram, spectral contrast feature, Tonnetz representation and zero-crossing rate. These features capture different aspects of the speech signal, such as timbre, pitch, energy, and harmony. We used a limited dataset of SER and augmented it with additional audios to increase the diversity and size of the data.

Th majority of past studies considered the performances of their models on a single dataset and achieved a good accuracy level. There could be several reasons why combining the CRE-MA-D, RAVDESS, Surrey, and Toronto datasets decreased the accuracy of our model, compared to when considering a single dataset. One possibility is that the datasets have different characteristics, such as different recording conditions, different speakers, or different emotional expressions. Combining datasets with different characteristics can introduce variability and noise into the data, which can make it more challenging for the model to accurately classify emotions.

Another possibility is that the model may have been biased in favor of a single dataset, when trained on that dataset alone. When multiple datasets are combined, the model may not be able to overfit to the characteristics of any single dataset, which could result in a decrease in accuracy.

We compared the performance of two models: one-dimensional convolutional neural network (conv1D) and random forest (RF) with RF-based feature selection. Conv1D is a deep learning model that can learn complex and nonlinear patterns from the audio features. On other hand, RF is a tree-based ensemble model that can handle high-dimensional and noisy data. RF-based feature selection is a technique that can reduce the dimensionality and redundancy of the data by selecting the most relevant features using another RF classifier.

Our results showed that RF with feature selection achieved higher average accuracy (69%) than conv1D (60%) and had the highest precision for fear (72%) and the highest recall for

calm (84%). This suggests that RF with feature selection can better discriminate between different emotions and capture the subtle variations in speech. Conv1D performed well for surprise (83% precision and recall), but poorly for calm (46% precision) and disgust (37% recall). This indicates that conv1D may have difficulty in distinguishing between low-intensity and negative emotions.

One possible reason for the superior performance of RF with feature selection is that it can reduce the noise and overfitting of the data by selecting about 100 features out of 174. Conv1D, on the other hand, may suffer from overfitting due to the limited size of the dataset and the complexity of the model. Another possible reason is that RF with feature selection can capture the nonlinear relationships between different features and emotions, while conv1D may be limited by the sequential nature of the input data.

Our study demonstrates the effectiveness of RF with feature selection for speech emotion classification using a limited dataset. However, there are some limitations and directions for future work. First, our dataset was relatively small and imbalanced, which may affect the generalization ability of our models. We augmented our dataset with additional audios, but more data from diverse sources and languages would be beneficial.

Second, our audio features were extracted manually, which may not capture all the relevant information from speech. We could explore other methods of feature extraction, such as deep learning or unsupervised learning. Third, our models were trained and tested on a single dataset, which may not reflect the real-world scenarios of SER. We could evaluate our models on other datasets or use cross-dataset validation to measure their robustness. Fourth, our models only used audio information for emotion recognition. We could incorporate other modalities, such as text or facial expressions, to improve the accuracy and reliability of SER.

## Author Contributions

**Conceptualization:** Mohammad Mahdi Rezapour Mashhadi.

**Data curation:** Mohammad Mahdi Rezapour Mashhadi, Kofi Osei-Bonsu.

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
