## [Decision Letter · Decision Letter 0]

9 Aug 2023

PONE-D-23-20208Speech Emotion Recognition using Machine Learning techniques: Feature Extraction and Comparison of Convolutional Neural Network and Random ForestPLOS ONE

Dear Dr. Rezapour Mashhadi,

Thank you for submitting your manuscript to PLOS ONE. After careful consideration, we feel that it has merit but does not fully meet PLOS ONE’s publication criteria as it currently stands. Therefore, we invite you to submit a revised version of the manuscript that addresses the points raised during the review process.

We look forward to receiving your revised manuscript.

Kind regards,

Felix Albu, Ph.D.

Academic Editor

PLOS ONE

Additional Editor Comments:

The decision is Major Revision.

Reviewers' comments:

Reviewer's Responses to Questions

**Comments to the Author**

1. Is the manuscript technically sound, and do the data support the conclusions?

Reviewer #1: Yes

Reviewer #2: Yes

2. Has the statistical analysis been performed appropriately and rigorously? 

Reviewer #1: N/A

Reviewer #2: Yes

3. Have the authors made all data underlying the findings in their manuscript fully available?

Reviewer #1: No

Reviewer #2: No

4. Is the manuscript presented in an intelligible fashion and written in standard English?

Reviewer #1: No

Reviewer #2: No

5. Review Comments to the Author

Reviewer #1: The authors have presented a machine learning based speech emotion classification method. There are few concerns on the work:

1) Emo-DB is an important dataset, which the authors have missed in their work.

2) The literature review is not adequate.

3) The results are not comparable to the state-of-the-art methods on the same dataset.

For a details of the literature and comparison, please refer to this article: https://arxiv.org/pdf/2112.05666.pdf or similar articles

Reviewer #2: 1. Please use single space tab for every para.

2. Represent abbreviations RAVDESS, TESS etc in caps in your paper.

3. Spectrographic representation of speech emotions are missing, represent them.

4. Graphical representation of results also missing as a comparision.

5. Need more to represent in flowchart.

6. PLOS authors have the option to publish the peer review history of their article (what does this mean?). If published, this will include your full peer review and any attached files.

Reviewer #1: No

Reviewer #2: **Yes: **P ANIL KUMAR

---

## [Decision Letter · Decision Letter 1]

31 Aug 2023

Speech Emotion Recognition using Machine Learning techniques: Feature Extraction and Comparison of Convolutional Neural Network and Random Forest

PONE-D-23-20208R1

Dear Dr. Rezapour Mashhadi,

We’re pleased to inform you that your manuscript has been judged scientifically suitable for publication and will be formally accepted for publication once it meets all outstanding technical requirements.

Kind regards,

Felix Albu, Ph.D.

Academic Editor

PLOS ONE

Additional Editor Comments (optional):

The decision is to Accept the paper.

Reviewers' comments:

Reviewer's Responses to Questions

**Comments to the Author**

1. If the authors have adequately addressed your comments raised in a previous round of review and you feel that this manuscript is now acceptable for publication, you may indicate that here to bypass the “Comments to the Author” section, enter your conflict of interest statement in the “Confidential to Editor” section, and submit your "Accept" recommendation.

Reviewer #1: All comments have been addressed

Reviewer #2: All comments have been addressed

2. Is the manuscript technically sound, and do the data support the conclusions?

Reviewer #1: Yes

Reviewer #2: Yes

3. Has the statistical analysis been performed appropriately and rigorously? 

Reviewer #1: Yes

Reviewer #2: Yes

4. Have the authors made all data underlying the findings in their manuscript fully available?

Reviewer #1: Yes

Reviewer #2: Yes

5. Is the manuscript presented in an intelligible fashion and written in standard English?

Reviewer #1: Yes

Reviewer #2: Yes

6. Review Comments to the Author

Reviewer #1: Authors have addressed the comments and substantially made the changes where necessary. Emo-DB dataset, though german, is used along with other datasets.

Reviewer #2: Ok , Please change the graphical representation colours , all appears in dark green .

Every thing what i have asked has covered in paper.

7. PLOS authors have the option to publish the peer review history of their article (what does this mean?). If published, this will include your full peer review and any attached files.

Reviewer #1: No

Reviewer #2: No

---

## [Editor Report · Acceptance letter]

4 Sep 2023

PONE-D-23-20208R1 

Speech Emotion Recognition using Machine Learning techniques: Feature Extraction and Comparison of Convolutional Neural Network and Random Forest 

Dear Dr. Rezapour Mashhadi:

I'm pleased to inform you that your manuscript has been deemed suitable for publication in PLOS ONE. Congratulations! Your manuscript is now with our production department. 

Kind regards, 

on behalf of

Dr. Felix Albu 

Academic Editor

PLOS ONE